# MHDNet: A Multi-Scale Hybrid Deep Learning Model for Person Re-Identification

**Jinghui Wang * and Jun Wang**

School of Mathematical Sciences, Jiangsu University, Zhenjiang 212013, China
* Correspondence: wangjinghui_ai@126.com

**Abstract:** The primary objective of person re-identification is to identify individuals from surveillance videos across various scenarios. Conventional pedestrian recognition models typically employ convolutional neural network (CNN) and vision transformer (ViT) networks to extract features, and while CNNs are adept at extracting local features through convolution operations, capturing global information can be challenging, especially when dealing with high-resolution images. In contrast, ViT rely on cascaded self-attention modules to capture long-range feature dependencies, sacrificing local feature details. In light of these limitations, this paper presents the MHDNet, a hybrid network structure for pedestrian recognition that combines convolutional operations and self-attention mechanisms to enhance representation learning. The MHDNet is built around the Feature Fusion Module (FFM), which harmonizes global and local features at different resolutions. With a parallel structure, the MHDNet model maximizes the preservation of local features and global representations. Experiments on two person re-identification datasets demonstrate the superiority of the MHDNet over other state-of-the-art methods.

**Keywords:** person Re-ID; deep learning; multi-scale feature; feature fusion module; MHDNet





## 1. Introduction

Person re-identification (Re-ID) is a critical field within computer vision that utilizes techniques to detect individuals in images or videos [1,2]. It can be considered a subfield within the domain of image retrieval. In recent years, Re-ID has emerged as a crucial research area due to its diverse applications in criminal investigations and computer vision. The primary goal of Re-ID is to efficiently retrieve and recognize specific individuals, thereby facilitating automatic identification in surveillance systems with non-overlapping fields across multiple cameras. However, challenges such as occlusion [3,4], illumination, human posture [5], and other confounding factors [6] can significantly impact the effectiveness of Re-ID in practical surveillance systems.

Early research on person re-identification focuses on extracting feature representations to distinguish individual identities [7,8] and designing an effective distance measurement to establish image similarity [9–11]. With the development of deep learning, the focus of early research on Re-ID has shifted towards developing robust feature representations [12–14] to distinguish individual identities, alongside the exploration of effective distance measurement techniques tailored to establish image similarity. Person re-identification based on deep learning can divide three aspects from the extracted image features: (1) Global feature learning is widely used in person re-identification. However, using attention mechanisms [15,16] to extract a global feature vector may cause the network to neglect local details, which are also important for accurate Re-ID. (2) Local feature learning is achieved by dividing the pedestrian image into smaller blocks to extract features from various parts and enhance the network's performance. However, using methods like grid segmentation and pose division can increase computational complexity, especially when using the pose division technique which may require an additional pose estimation network to improve performance. (3) Integrating global and local features through fusion learning [13,17] can help

the network capture both global and local information to improve the identification of pedestrians [18–20]. Although this approach leads to better semantic feature extraction, it can increase the model's complexity and network parameters. Additionally, it is important to note that current methods mainly concentrate on deep semantic features and may not consider other relevant factors.

Current research focuses primarily on the output features of high layers, overlooking the learning potential of shallow components for detecting subtle details of a person. Furthermore, the use of multi-loss joint training enables evaluation of person feature distances across several dimensions, while multi-loss approaches have potential, it is important to note that different losses have varying impacts on results, necessitating the incorporation of novel loss combinations. However, utilizing multiple losses increases training costs significantly as the computational complexity of the system is augmented.

To address the above-mentioned issues, this paper proposes a multi-scale interactive transformer structure that utilizes Vision Transformer as the Re-ID backbone to extract hierarchical information and fine-grained features. Additionally, a multi-scale feature fusion module is proposed to achieve bidirectional fusion of the global modeling ability of the Transformer and the local capture ability of the CNN, effectively improving the performance and stability of the model. The joint action of the two modules improves the network's perception of crucial person information. This paper also incorporates a learnable attention mechanism to further enhance the model's performance by allocating more attention to features. Experimental results on widely used datasets demonstrate that the method proposed in this paper achieves impressive performance and robustness in Re-ID tasks, significantly improving mAP and Top-1 accuracy compared to the current state-of-the-art methods. The primary contributions of this paper are as follows:

(1) We investigated the feasibility of replacing convolutional neural networks (CNNs) with vision transformers (ViT) in the person re-identification (Re-ID) domain, and we successfully integrated visual specificity induction bias into the standard ViT architecture, achieving comparable and enhanced performance comparable to recent transformer-based models.

(2) We introduce a spatial prior module and two feature interaction operations, seamlessly integrating them into the backbone network in a hybrid manner. This approach harmonizes the strengths of various modules, fostering efficient information exchange within the network. As a result, our model captures local details more precisely and re-organizes fine-grained multi-scale features, significantly improving matching accuracy and stability.

(3) We conducted rigorous experiments and ablation studies on benchmark single-person Re-ID datasets, Market1501 and DukeMTMC-reID, along with cross-dataset validation to assess our method's generalization. Our approach's effectiveness is evident from numerous experiments, demonstrating competitive performance against state-of-the-art methods.

## 2. Related Work

In this section, we will review previous relevant work to provide reference to our research and compare and analyze our work with these studies.

### 2.1. Transformer in Person Re-ID

In recent years, with the widespread use of Transformer [21] in natural processing, research on its application to computer vision tasks [22–24] to study image dependencies has gained momentum. In Re-ID [1,2,25], spatial alignment [13,26] and multi-scale features [27,28] are crucial for feature learning. However, implementing spatial alignment and multi-scale features in visual-based Transformer structure remains a challenging research area. In their work, He et al. [29] proposed a side information embedding and puzzle patch modules to discriminative features in a pure Transformer framework for improving performance. Chen et al. [30] constructed fully relational features for Re-ID with a fully relational high-order transformer structure (OH Transformer), acquiring attention matrices based on queries and isolated key pairs at each spatial position and further modeling

high-order statistical information for non-local mechanisms. To enhance the robustness of feature extraction, Chen et al. [31] designed a hybrid backbone Res Transformer by combining ResNet-50 with Transformer blocks. Lai et al. [32] proposed an adaptive component partitioning model to extract local features more effectively in human Re-ID. In this paper, we propose a bidirectional fusion of Transformer's global modeling ability and CNN's local capture ability. This is achieved by combining Transformer with CNN, and we utilize a multi-scale feature fusion module for feature extraction at different levels. Our proposed approach achieves state-of-the-art performance on three datasets.

### 2.2. Multi-Scale Feature Method

In Re-ID tasks, different feature levels such as color, texture, shape, and posture contribute to feature learning. Therefore, introducing multi-scale features has shown potential in improving model performance. The method of multi-scale features involves adding various feature extraction branches, each with a different scale, to the backbone network. The features from different scales extracted by these branches are aggregated for training, as shown in Figure 1d. This approach aims to learn the depth features of different body layers, optimizing model performance. Cai et al. [33] used attention mechanisms to extract characteristics at three scales. Wang et al. [34] extracted multi-scale features based on different stages of the backbone network and added these to the subsequent coding task. Liu et al. [35] proposed a multi-scale feature enhancement model that aggregates shallow, medium, and deep-level features to extract feature maps with good spatial structures and rich semantic information. Wu et al. [27] proposed an attention depth framework with multi-scale deep supervision by inserting attention modules into the backbone network for efficient feature extraction. Zhou et al. [36] proposed a lightweight full-scale network model that extracts multi-scale features through full-scale residual blocks. Chen et al. used feature block processing and multi-layer fusion methods to extract multi-scale features. Hao Zhang et al. [37] proposed the multi-scale visual attribute co-attention model (mVACA), enhancing zero-shot image recognition performance and achieving outstanding results in benchmarks. Zhang et al. [38] developed an innovative region-based multi-scale network that significantly improves emotional image recognition by seamlessly integrating features extracted from focal regions along with their surrounding contexts. Zhou et al. [39] introduced a DNN model that integrates spatial pyramid pooling with feature pyramids to enhance distorted image quality assessment. This model leverages deep end-to-end supervision to efficiently utilize multi-scale features, thereby improving the accuracy of perceptual quality prediction. Chen et al. [40] introduced an innovative multi-scale SER parallel network, AMSNet, which integrates fine-grained frame-level manual features with coarse-grained utterance-level deep features. This network also incorporates an enhanced attention-based LSTM and a CNN with Squeeze-and-Excitation blocks (SCNN) to extract emotional information from speech signals more comprehensively. Hu et al. [41] innovatively introduced the Multi-scale Multi-angle Attention Network (MMAN), integrating 3D and 2D convolutional layers to extract spectral and spatial features, respectively, for a comprehensive understanding of image content. While these methods have their advantages and disadvantages, they effectively extract feature information at various scales and prove to be beneficial attempts and explorations for pedestrian recognition tasks.

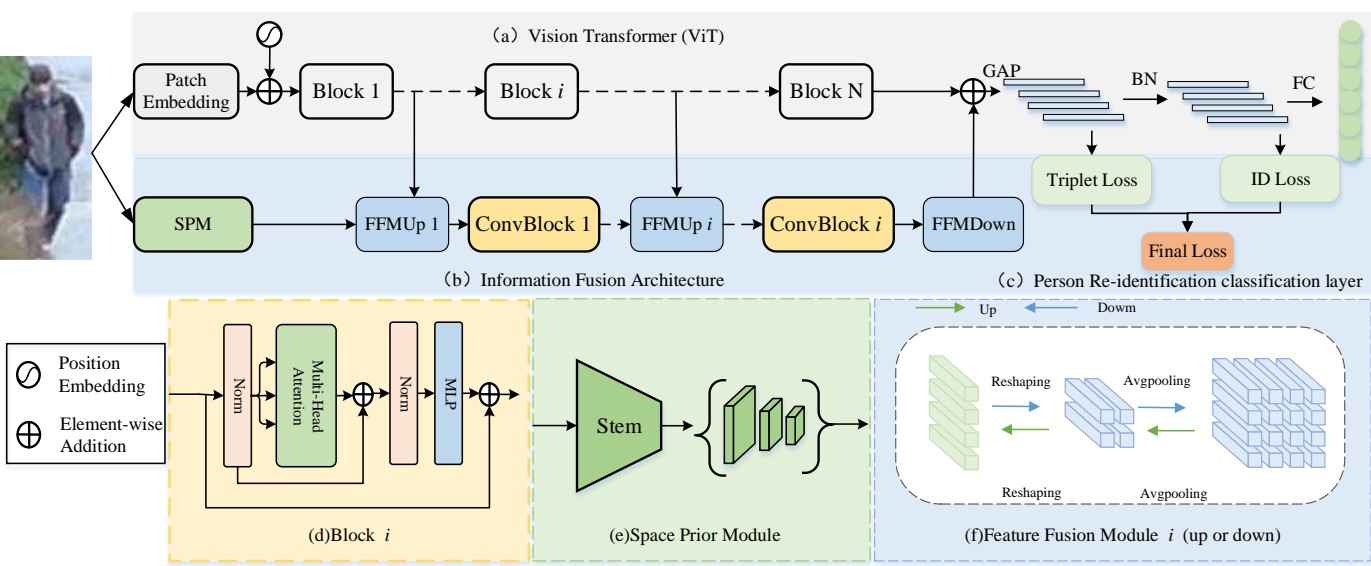

**Figure 1.** Overall architecture of MHDNet. (**a**) The ViT, whose encoder layers are divided into N (usually N = 12) equal blocks (**c**) for feature interaction. (**b**) Our Information Fusion Architecture, which contains two key designs, including (**e**) a spatial prior module for modeling local spatial contexts from the input image, (**f**) Feature Fusion Module for introducing spatial priors into ViT. (**d**) Block *i*.

## 3. The Proposed Method

This section provides a comprehensive exposition of our novel approach that addresses the research problem.

### 3.1. Framework Overview

In order to tackle the intricacies of the detection framework and the absence of dimensional correlation inherent in the domain of person re-identification, we present our network architecture. The proposed person re-ID architecture based on multi-scale hybrid features includes a ViT architecture, an information fusion architecture consisting of a spatial prior module and a feature fusion module, and a person re-identification classification layer.

As illustrated in Figure 1, we focus on optimizing both the network model architecture and the deep model. This paper employs a modified Vision Transformer as our backbone for generating person identity features. The network structure is shown in Figure 1a, consisting of a patch embedding layer and N Transformer encoder layers. We employ patch embedding technology to partition the image into $16 \times 16$ non-overlapping small blocks. These blocks are then flattened and mapped to the feature space, resulting in a reduction of feature resolution by a factor of 16 compared to the original image. Following this, we apply positional encoding to incorporate spatial relationships between the features before feeding them into $N$ encoder layers for further processing. Within each encoder layer, every feature participates in a global attention mechanism that enables it to interact with all other features for information exchange.

Regarding information fusion architecture, this paper employs an efficient and innovative approach, as illustrated in Figure 1b. We process the input image by feeding it into the spatial prior module to obtain three spatial features at different scales. Following this, we flatten these feature maps as inputs for feature interaction. Specifically, we divide the ViT encoder into $N$ blocks, where $N$ is typically 12. For each block, we inject the spatial prior information into it through the spatial feature fusion module and then extract hierarchical features from the output, using a multi-scale feature extractor. Next, we perform $N$ feature interactions to obtain high-quality multi-scale features for enhancing the model's performance.

The person re-identification classification layer of the network includes a linear layer, a normalization layer, and a fully connected layer, as illustrated in Figure 1c. We transform the input feature vector into a 768-dimensional feature representation for person identity description. Subsequently, we determine the person's identity in the re-identification task using a fully connected layer.

### 3.2. Space Prior Module

Recent studies [42,43] have shown that incorporating convolution operations into the Transformer architecture can better capture local spatial information. However, rather than modifying the original ViT architecture, we were inspired to introduce the Spatial Prior Module ($SPM$) to address this issue. The $SPM$ module is designed to work parallel with the embedding layer while representing the local spatial context of the input image. Its precise definition is articulated as follows:

$$F_1, F_2, F_3 = SPM(X) \tag{1}$$

where $X \in R^{(C \times H \times W)}$ represents the holistic depiction of the input image.

As illustrated in Figure 1d, the proposed module includes three convolutional layers and one maximum pooling layer to extract and downsample features from the input image. We then utilize a set of $1 \times 1$ strided convolutions to expand the number of channels four times without altering the feature map size. This approach yields a feature pyramid $\{F_1, F_2, F_3\}$ which is used as an input for feature interaction. The precise computational procedure for $F_i$ is as follows.

$$F_{i+1} = f(\text{BN}(Conv(F_i, w_i), \beta_i, \gamma_i)) \tag{2}$$

where $f$ represents the activation function (such as $ReLU$), $Conv$ denotes the convolution operation, and $BN$ denotes the batch operation. $w$ represents the weight of convolution kernel, and $\beta$ and $\gamma$ represent the scale and factors in batch normalization operation.

### 3.3. Feature Fusion Module

In order to tackle the intricacies of the detection framework and the absence of dimensional correlation inherent, we introduce the Global Feature Fusion Module ($FFM$), as illustrated in Figure 2. This module enables the continuous coupling of local features and global representations via interaction.

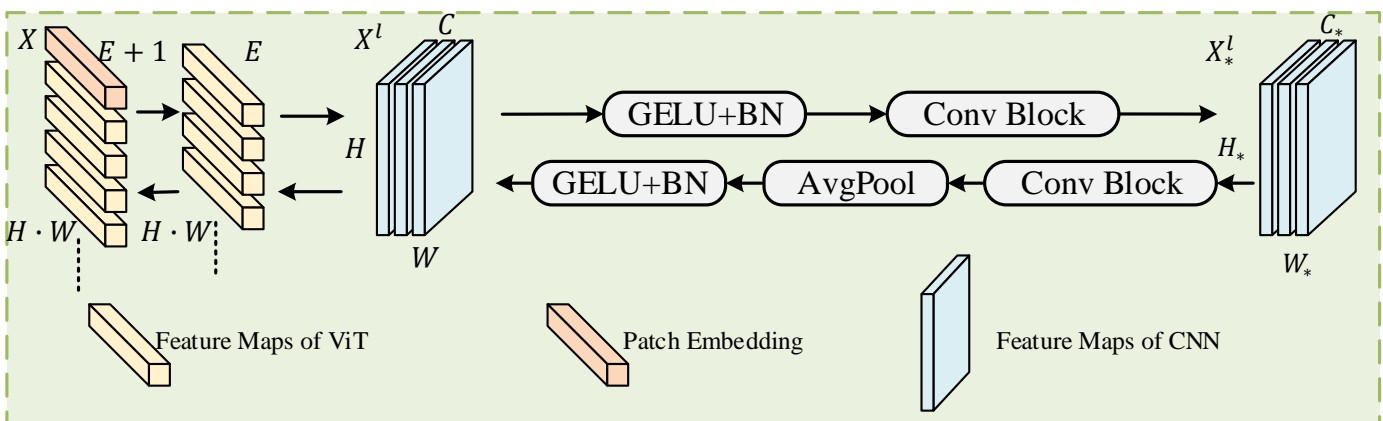

**Figure 2.** Feature fusion module.

It is important to note that CNN and Transformer models usually have different feature dimensions. In CNN, the feature map has dimensions of $F \in R^{C \times H \times W}$, where $C$, $H$, and $W$ represent the number of channels, height, and width, respectively. In Transformer,

the shape of the patch embedding is $P \in R^{(K+1) \times E}$, where $K$, 1, and $E$ represent the number of multiple patches, tokens, and embedded dimensions of the image, respectively.

To ensure the alignment of spatial dimensions between the feature map in the CNN branch and the patch embedding in the Transformer branch, an upsampling operation is performed on the CNN feature map $F^s \in R^{(C^s \times H^s \times W^s)}$. The feature map is then adjusted to $C^s$ channels using a $14 \times 1$ convolution to obtain $F^s_{conv} \in R^{(C^s \times H^s \times W^s)}$. Finally, the upsampled feature map is added to $F$ as $F_{final} = F + F^s_{conv}$.

When propagating from the branch to the Transformer branch, a $1 \times 1$ convolution operation is applied to adjust the channel number to $C^t$, resulting in $P^t_{conv} \in R^{C^t \times (K+1) \times E}$. The downsampling operation is then used to align the spatial dimension, obtaining $P^t_{convpool} \in R^{(C^t \times (K+p) \times E)}$, where $p$ represents the number of spatial dimensions that need to be expanded. Add $P^t_{convpool}$ to patch embedding $P$ to obtain $P_{final} = P + P^t_{conv} \in R^{((K+p) \times E)}$. The final $F_{final}$ and $P_{final}$ are normalized by using LayerNorm and BatchNorm.

In contrast, feature maps and patch embeddings exhibit significant semantic differences. Local convolution operators identify features within a local region and encode them into distinct feature maps. Conversely, the global self-attention mechanism considers all information and captures global relationships between pixels. To bridge these semantic differences, we introduce $FFM$ to each block (except the first) in order to progressively integrate semantic information across the network.

### 3.4. Loss Function

In person re-identification tasks, the loss function plays a crucial role in model training by adjusting the model's discriminative ability and capturing correlations between similar samples, either individually or jointly. To comprehensively evaluate the proposed model under different loss constraints in person re-identification (Re-ID) tasks, incorporate several metric learning loss functions. These include the ID loss, triplet loss, circle loss, contrast loss, instance loss, and sphere loss, which can be used individually or in combination.

3.4.1. Cross-Entropy Loss

In person re-identification tasks, the cross-entropy loss is the most commonly used ID loss and is used to measure the difference between the results of the model and the actual labels. The smaller the cross-entropy loss, the closer the predicted results and actual labels are.

Specifically, for an input image $x$, the predicted output of the model for class $i$ is denoted as $p_i$ (the prediction probability for class $i$), and the true label of $x$ is denoted as $y$. The expression for the cross-entropy loss is given by Equation (3):

$$L_{CE} = \sum_i^N -q_i \log(p_i) \begin{cases} q_i = 0, y \neq i \\ q_i = 1, y = i \end{cases} \tag{3}$$

In ReID tasks, noise, missing labels, or inaccurate labels can lead to overfitting of the model and limit its general ability when using a separate cross-entropy loss. To address this issue, we propose the use of label smoothing [44] in cross-entropy to prevent the loss of negative samples from being ignored. Unlike traditional cross-entropy loss, the class label $q_i$ is calculated using a certain probability instead of being strictly defined as either 0 or 1. This improves the model's performance, and $\alpha$ is represented by Equation (4):

$$q_i = \begin{cases} 1 - \frac{N-1}{N}\varepsilon, & if \quad i = y \\ \frac{\varepsilon}{N}, & otherwise \end{cases} \tag{4}$$

where $N$ represents the number of classes in the multi-classification problem, and $\varepsilon$ is a small hyperparameter.

### 3.4.2. Triplet Loss

Person re-identification tasks present unique challenges in terms of training models to accurately measure distance and ensure high discriminability, making traditional loss functions (such as the cross-entropy loss) inadequate. To address this, we selected metric-based learning using the triplet loss function [45]. By using the distance between similar and dissimilar samples to optimize the feature embedding space, triplet loss compensates for the limitations of simple metric learning. The formula for calculating triplet loss is shown in Equation (5):

$$L_{Tri} = \lfloor d_p - d_n - \alpha \rfloor_+ \tag{5}$$

where $d_p$ and $d_n$ represents the distance between the feature representation pairs of positive and negative sample pairs, usually calculated using Euclidean distance or cosine distance. $\alpha$ is the marginal distance of loss.

### 3.4.3. Circle Loss

Referring to Equations (3) and (5), the circles of the same color show person features of the same identity. The identity loss calculates the cosine distance between features, while the triplet loss calculates the Euclidean distance between features. In person Re-ID tasks, using identity loss alone for metric learning generally achieves better results than triplet loss, and we also validate this claim. The Circle loss can be defined as follows, where a given pedestrian sample $x$ in the feature space has $K$ within-class similarity scores and $L$ between-class similarity scores:

$$L_{Circle} = \log \left[ 1 + \sum_{j=1}^{L} \exp\left( \gamma \alpha_n^j \left( s_n^j - \Delta n \right) \right) \sum_{i=1}^{K} \exp\left( -\gamma \alpha_p^i \left( s_p^i - \Delta p \right) \right) \right] \tag{6}$$

where $s_n^j$ represent between-class similarity, $j \in \{1, 2, \ldots, L\}$. $s_p^i$ represent within-class similarity, $i \in \{1, 2, \ldots, K\}$. $\gamma$ is the scale factor. $\alpha_n^j$ and $\alpha_p^i$ are linear factors that constrain the learning rate of $s_n^j$ and $s_p^i$. $\Delta n$ and $\Delta p$ denote the between-class and within-class thresholds, respectively.

## 4. Experiment

In this section, we conducted experiments to verify the proposed method's effectiveness and compare it with the state-of-the-art techniques. We selected two benchmark Re-ID datasets and conducted comparative experiments using the suggested method, as presented in Table 1.

**Table 1.** Person Re-ID datasets introduction.

| Datasets | Cameras | TrainIDs | TrainImgs | TestIDs | QueryImgs | GalleryImgs |
|---|---|---|---|---|---|---|
| Market-1501 [46] | 6 | 751 | 12,396 | 750 | 3368 | 19,732 |
| DukeMTMC-reID [36] | 8 | 702 | 16,522 | 702 | 2228 | 17,661 |

### 4.1. Datasets

Market1501 [46] dataset comprises 32,668 images of 1501 individuals captured by six cameras. Each individual is captured by at least two cameras, and the dataset includes multiple training image sets of 751 individuals, totaling 12,936 images and averaging 17.2 training images per person. The test set contains 19,732 images of 750 individuals, with 3368 of these images used for query purposes, and the remaining images used as a library.

DukeMTMC-ReID [36] dataset comprises 36,411 images and 1812 identities captured by eight different cameras. Of the identities, 1404 were captured by more than one camera, while 408 were captured by only one camera. The training set includes 702 images of identities, while the remaining identities were used for testing. For each identity in each

camera, one image was selected as the query set in the test set, while the rest of the images were reserved as the image library.

Evaluation indicators: Two commonly used metrics for pedestrian recognition tasks are mean average precision (mAP) and cumulative characteristic (CMC). The traditional accuracy indicator, referred to as Top-1 accuracy, reflects the matching result between the identity with the highest probability prediction by the model and the ground truth. Conversely, Top-5 accuracy is determined by considering the identities with the top five highest probability predictions.

### 4.2. Implementation Details

The experiments in this article were conducted on a Windows 11 operating system with a 12th generation Intel Core i9-12900H processor and 30 GB of memory, and an Nvidia RTX 3090 graphics card (24 GB) using CUDA version 11.6. We used Python 3.8.17 and PyTorch 1.12.1 for data processing.

The model was trained for 500 epochs with images resized to 256 × 128. Stochastic Gradient Descent (SGD) was utilized as the optimization algorithm with the learning rate initialized to 0.08 and subsequently optimized using a simulated annealing algorithm. We set the drop rate to 0.1 to minimize overfitting.

### 4.3. Ablation Study

The effectiveness of SPM and FFM. To corroborate the performance advantages of SPM and FFM, we compare the performance of different backbones on MHDNet in Table 2 and Figure 3. We can observe that ViT series backbones bring better retrieval performance, but at the same time, their increased model complexity brings more inference time consumption. It is important to highlight that in the Market-1501, MHDNet achieves the mAP of 87.7, which is 10.2 higher than the value of 77.5 by ResNet-50. Similarly, in the DukeMTMCID, MHDNet obtains the mAP value of 79.7, which is 11.2 higher than the 68.5 mAP value of ResNet-50. This outcome underlines that MHDNet leveraging SPM and FFM is significantly superior to other methods on two datasets. Our experiments have demonstrated that the SPM and FFM components have a considerable impact on improving the baseline performance.

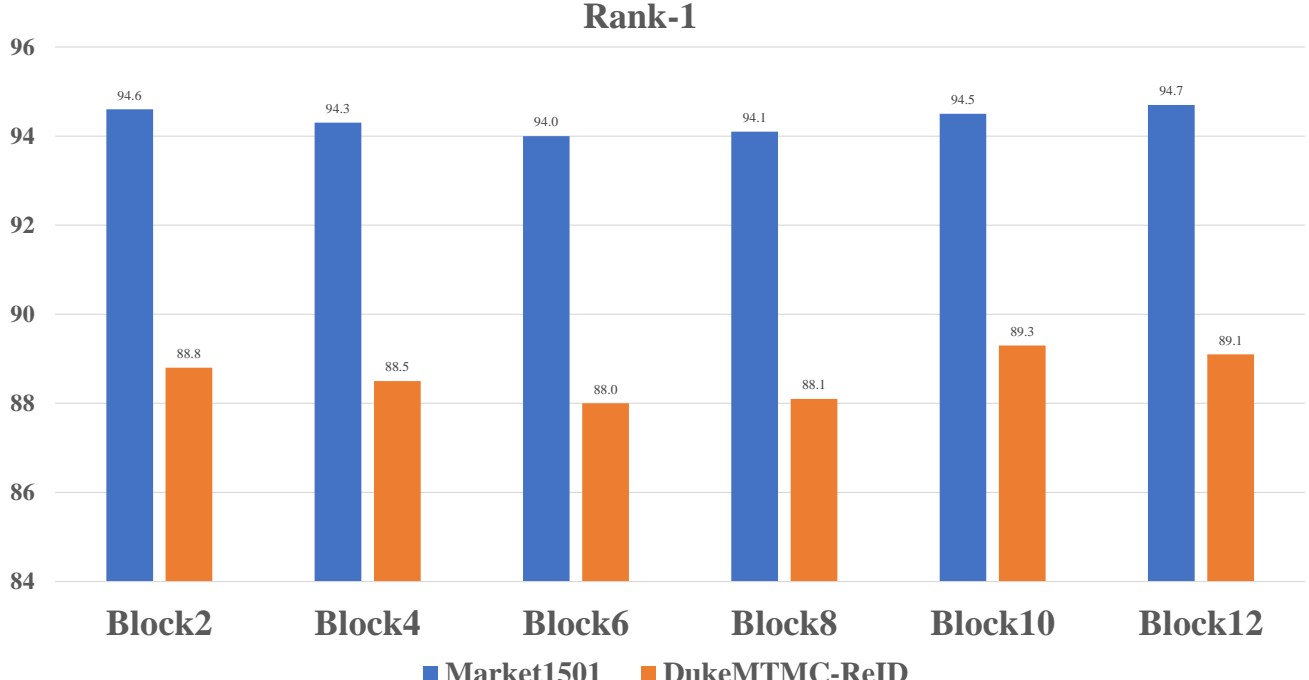

**Figure 3.** Evaluate the Rank-1 of effective placement of Block on two datasets.

**Table 2.** Comparison of different backbones.

| Methods | Inference Time | Market1501 | | | DukeMTMC-reID | | |
|---|---|---|---|---|---|---|---|
| | | mAP | Rank 1 | Rank 5 | mAP | Rank 1 | Rank 5 |
| ResNet-18 | 5 ms | 77.5 | 91.2 | 95.5 | 68.5 | 82.2 | 92.8 |
| ResNet-34 | 11 ms | 82.4 | 92.5 | 96.2 | 73.4 | 84.7 | 93.4 |
| ResNet-50 | 16 ms | 85.7 | 94.2 | 97.6 | 75.9 | 86.2 | 94.7 |
| ResNet-101 | 24 ms | 86.8 | 94.4 | 98.3 | 77.2 | 87.3 | 95.2 |
| ViT-B | 40 ms | 86.8 | 94.5 | 98.1 | 79.3 | 88.8 | 95.4 |
| Deit-B | 36 ms | 86.6 | 94.4 | 97.8 | 78.9 | 88.3 | 95.1 |
| MHDNet (Our) | 38 ms | 87.7 | 94.6 | 98.6 | 79.7 | 89.1 | 95.6 |

Effective Feature Selection. In order to demonstrate the effectiveness of each module in the proposed MHDNet, we experiment on Market-1501, DukeMTMC-ReID. We remove all the above modules and set the ViT (ViT with basic operation module) as the baseline. Starting with the baseline, three modules are added to the baseline model in turn. The experimental results are shown in Table 3 and Figure 4. In Table 3, we found using GAP in MHDNet performed better than baseline, while BN did not yield significant improvements until BN was introduced. In addition, we experimented with the effect of MHDNet with an FC layer and found that its accuracy is not as high as MHDNet with one-dimensional convolution.

**Table 3.** Ablation studies of the proposed method on individual components.

| Methods | Market1501 | | | | DukeMTMC-reID | | | |
|---|---|---|---|---|---|---|---|---|
| | mAP | Rank 1 | Rank 5 | Rank 10 | mAP | Rank 1 | Rank 5 | Rank 10 |
| Baseline | 84.0 | 92.4 | 95.4 | 96.6 | 76.4 | 85.4 | 93.2 | 94.4 |
| + GAP | 85.1 | 93.7 | 97.2 | 98.0 | 77.8 | 87.1 | 94.6 | 95.3 |
| + BN | 85.2 | 93.9 | 97.5 | 98.1 | 78.1 | 87.5 | 94.9 | 95.5 |
| + FC Layer | 86.8 | 94.5 | 98.1 | 98.9 | 79.3 | 88.8 | 95.4 | 96.2 |
| MHDNet (Our) | 87.7 | 94.6 | 98.6 | 99.3 | 79.7 | 89.1 | 95.6 | 96.6 |

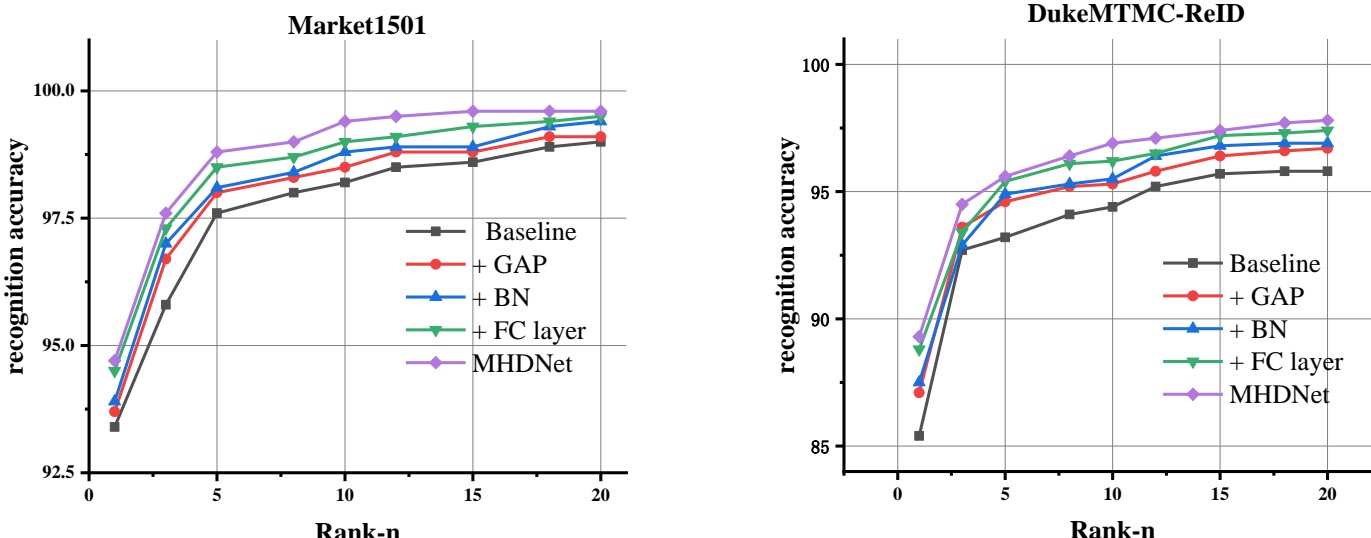

**Figure 4.** CMC curve of Baseline, Baseline + GAP, Baseline + GAP + BN, MHDNet with FC Layer and MHDNet on the two datasets.

### 4.4. Effectiveness of Different Loss Functions

Since different loss functions have other effects on person re-identificatin features, the model performance is also significantly different. So, we conducted ablation experiments based on loss selection to demonstrate the differential effects of different loss functions, as shown in Tables 4 and 5.

Based on empirical evidence, the ID loss outperforms the Triplet loss when both losses are used separately. Moreover, when combined with the Triplet loss or Circle loss, the ID loss demonstrates superior performance compared to the Instance loss and Sphere loss. This indicates that sensible choice of loss function for metric learning can consistently enhance model performance. However, considering our primary focus on showcasing the performance of the FFM module, we limit our subsequent experiments to using only the ID loss and Triplet loss as the distance metrics.

**Table 4.** The effectiveness of loss selections on Market1501.

| Loss Function | mAP | Rank 1 | Rank 5 | Rank 10 |
|---|---|---|---|---|
| ID | 81.8 | 93.1 | 97.2 | 98.5 |
| Triplet | 80.3 | 91.8 | 96.2 | 98.1 |
| ID + Spere | 78.5 | 90.9 | 97.1 | 98.2 |
| ID + Contrast | 82.6 | 93.0 | 97.9 | 98.7 |
| ID + Cricle | 83.4 | 93.7 | 98.1 | 98.9 |
| ID + Instance | 82.2 | 92.9 | 97.7 | 98.6 |
| ID + Triplet | 87.7 | 94.6 | 98.6 | 99.3 |
| ID + Triplet + Cricle | 84.1 | 93.6 | 98.3 | 98.8 |
| ID + Triplet + Instance | 83.8 | 93.4 | 97.5 | 98.9 |

**Table 5.** The effectiveness of loss selections on DukeMTMC-reID.

| Loss Function | mAP | Rank 1 | Rank 5 | Rank 10 |
|---|---|---|---|---|
| ID | 75.3 | 86.4 | 94.5 | 96.2 |
| Triplet | 74.2 | 85.8 | 93.7 | 96.1 |
| ID + Spere | 71.8 | 81.6 | 94.1 | 95.9 |
| ID + Contrast | 76.9 | 87.8 | 95.2 | 96.3 |
| ID + Cricle | 77.6 | 88.1 | 95.4 | 96.5 |
| ID + Instance | 76.2 | 87.6 | 95.1 | 96.0 |
| ID + Triplet | 79.7 | 89.1 | 95.6 | 96.6 |
| ID + Triplet + Cricle | 78.5 | 88.7 | 95.5 | 96.6 |
| ID + Triplet + Instance | 77.4 | 87.9 | 95.0 | 96.4 |

### 4.5. Comparison with the State-of-the-Art Methods in Single-Domain Person Re-Identification

In this Section, we compare MHDNet with other methods on Market-1501 [46] and DukeMTMC-ReID [36], as shown in Table 6. For fair comparisons, no post-processing such as re-ranking strategies or multi-query fusion was used for our methods.

Table 6 shows that our MHDNet achieved 87.7% mAP and 94.6% Rank-1 on the Market-1501 [46]. Its Rank-1 accuracy is slightly, 0.8%, lower than PAT and the same as Rank1 accuracy in TransReID, yet MHDNet clearly surpasses all methods in terms of mAP. On the DukeMTMC-ReID [36] dataset, MHDNet achieved the Rank-1 accuracy of 89.1% and the mAP value of 79.7%, respectively, which are higher than most models. However, it still cannot reach the level of some recent excellent models in all indicators. This is because we only improved the model's performance by modifying the backbone to make it work well on devices with limited performance, while strictly limiting the model's resource consumption. In contrast, other models do not have this limitation.

As a result, our model achieves results comparable to most classical models, but there is still a gap compared to the best models. Compared with other models, we improve only on the backbone to improve the accuracy, which has better scalability and can be easily combined with other methods to further improve the model accuracy.

**Table 6.** Validation results on Market1501 and DukeMTMC-ReID. Bold indicates the best performance.

| Method | Venue | Market1501 | | DukeMTMC-reID | |
|---|---|---|---|---|---|
| | | mAP | Rank 1 | mAP | Rank 1 |
| PCB [47] | ECCV 2018 | 81.6 | 93.8 | 69.2 | 83.3 |
| BoT [14] | CVPR 2019 | 85.9 | 94.5 | 76.4 | 86.4 |
| HOReID [26] | CVPR 2020 | 84.9 | 94.2 | 75.6 | 86.9 |
| M-DEFNet | MTA 2020 | 82.7 | 94.8 | 73.1 | 84.7 |
| ADC-2OIB [48] | CVPR 2021 | 87.7 | 94.8 | 74.9 | 87.4 |
| DAReID [49] | KBS 2021 | 87.0 | 94.6 | 78.4 | 88.9 |
| OSNet [50] | TPAMI 2021 | 86.7 | 94.8 | 76.3 | 88.7 |
| ASAN [51] | TCSVT 2021 | 85.3 | 94.6 | 76.3 | 88.7 |
| CDNet [52] | CVPR 2021 | 86.0 | 95.1 | 76.8 | 88.6 |
| PAT [53] | CVPR 2021 | 86.6 | 95.4 | 78.2 | 88.8 |
| L3DS [54] | CVPR 2021 | 87.3 | 95.0 | 76.1 | 88.2 |
| TransReID [29] | ICCV 2021 | 86.8 | 94.6 | 79.3 | 88.8 |
| ConRFL [55] | PR 2022 | 81.4 | 92.8 | 68.4 | 80.5 |
| CAL | CVPR 2022 | 87.5 | 94.7 | 74.1 | 86.2 |
| AOPS [51] | TCSVT 2022 | 84.1 | 93.4 | 74.1 | 86.2 |
| DeiT-Small + DCAL [56] | CVPR 2022 | 85.3 | 94.0 | 77.4 | 87.9 |
| IIANet | MTA 2023 | 84.9 | 94.2 | - | - |
| With Res2Net50 [57] | Sensors 2023 | 87.1 | 95.0 | 77.6 | 88.1 |
| DWNet-R [58] | Sensors 2023 | 87.5 | 94.9 | 79.1 | 88.4 |
| UV-ReID-ABLM | MVA 2023 | 75.0 | 89.9 | 61.8 | 81.4 |
| ICAM [59] | EAAI 2023 | 82.3 | 93.3 | 71.6 | 85.6 |
| MHDNet (Ours) | | 87.7 | 94.6 | 79.7 | 89.1 |

*4.6. Comparison with the State-of-the-Art Methods in Cross-Domain Person Re-Identification*

To assess and demonstrate the efficacy and superiority of MHDNet, we conducted a comparative evaluation of the proposed approach against several leading UDA re-ID methods on the Market-1501→DukeMTMC and DukeMTMC→Market-1501. The experimental outcomes are summarized in Table 7, providing quantitative evidence for the performance of MHDNet in UDA re-ID scenarios.

**Table 7.** Validation results on M→D and D→M. Bold indicates the best performance.

| Method | DukeMTMC-reID→Market1501 | | | | Market1501→DukeMTMC-reID | | | |
|---|---|---|---|---|---|---|---|---|
| | mAP | Rank 1 | Rank 5 | Rank 10 | mAP | Rank 1 | Rank 5 | Rank 10 |
| CFSM [60] | 28.3 | 61.2 | - | - | 27.3 | 49.8 | - | - |
| UCDA-CCE [61] | 30.9 | 60.4 | - | - | 31.0 | 47.7 | - | - |
| UTAL [62] | 46.2 | 69.2 | - | - | 44.6 | 62.3 | - | - |
| ECN [63] | 43.0 | 75.6 | 87.5 | 91.6 | 40.4 | 63.3 | 75.8 | 80.4 |
| PDA-Net [64] | 47.6 | 75.2 | 86.3 | 90.2 | 45.1 | 63.2 | 77.0 | 82.5 |
| CR-CAN+ [65] | 54.0 | 77.7 | 89.7 | 92.7 | 48.6 | 68.9 | 80.2 | 84.7 |
| D-MMD [66] | 48.8 | 70.6 | 87.0 | 91.5 | 46.0 | 63.5 | 78.8 | 83.9 |
| AD-Cluster [67] | 68.3 | 86.7 | 94.4 | 96.5 | 54.1 | 72.6 | 82.5 | 85.5 |
| CGAN-TM [68] | 35.2 | 57.3 | - | - | 36.2 | 65.3 | - | - |
| Soft-mask [69] | 69.5 | 86.9 | - | - | 61.3 | 76.9 | - | - |
| PREST [70] | 62.4 | 82.5 | 92.1 | 94.9 | 56.1 | 74.4 | 83.7 | 85.9 |
| CAC–CSP [71] | 36.9 | 69.4 | 82.8 | - | 37.0 | 57.5 | 71.2 | - |
| EDAAN [72] | 35.4 | 64.5 | 83.0 | - | 39.6 | 57.8 | 72.2 | - |
| 3D-GAT [73] | 28.6 | 59.4 | 75.2 | - | 26.1 | 45.1 | 59.3 | - |
| STReID [74] | 31.6 | 62.3 | 79.1 | - | 29.2 | 52.3 | 65.9 | - |
| UADA-SD [75] | 30.2 | 57.4 | 72.4 | 30.3 | 45.3 | 57.8 | - | |
| MHDNet (Ours) | 63.8 | 81.8 | 88.8 | 91.5 | 58.8 | 70.9 | 80.8 | 83.4 |

From Table 7, we can observe that MHDNet achieves 83.8% Rank-1 and 63.8% mAP on the DukeMTMC→Market-1501 and 76.4% and 58.8% on the Market-1501→DukeMTMC, respectively. Specifically, compared to PREST, which employs a progressive representation

enhancement approach, MHDNet demonstrates improvements of 2.0% and 2.7% in Rank-1 and mAP on the Market-1501→DukeMTMC and 1.3% and 1.4% on the DukeMTMC→Market-1501, respectively. This suggests that MHDNet effectively reduces the domain gap between the source and target domains and gradually adapts to the data distributions of both domains compared to PREST. In comparison to Soft-mask, the top performer in Table 7, MHDNet demonstrates a minor decrement in performance, achieving a slightly lower mAP by 1.7% and a Rank-1 accuracy reduction of 2.1% on the DukeMTMC→Market-1501. Similarly, on the Market-1501→DukeMTMC, MHDNet exhibits a minor decline in Rank-1 accuracy by 0.5% and a 2.5% drop in mAP. Evidently, MHDNet does not possess a notable edge over Soft-mask in the DukeMTMC→Market-1501 and Market-1501→DukeMTMC. This could be due to the significant stylistic disparities between the Market-1501 and DukeMTMC datasets, originating from Asia and America, respectively, as well as the limitations in our network's cross-domain fusion capabilities. In our future endeavors, we aim to investigate strategies that effectively integrate domains exhibiting substantial stylistic variations.

### 4.7. Visualized Attention Maps of the MHDNet

In this section, we conduct an exploratory analysis of the performance of the proposed network by visualizing the final output feature maps of MHDNet, ResNet-50, DeiT, and ViT, as shown in Figure 5. The purpose of this analysis is to gain a deeper understanding of the characteristics of the network and to evaluate whether our design has achieved the expected performance.

It is evident from the first and second rows on the left side of the attached image that ResNet-50 struggles in handling regional features, whereas MHDNet demonstrates profound attention to such features, excelling not only in balance but also in notable multi-scale attention capabilities. In contrast, the first and second rows on the right side reveal that while the traditional ViT structure model can, to a certain extent, focus on regional features, it falls short in extracting these features with precision. MHDNet achieves remarkable balance between local and global information, enabling accurate separation of individuals from their surroundings. These characteristics fully demonstrate that the use of MHDNet in person re-identification (ReID) tasks can more effectively capture human body features, enhance recognition accuracy, and better adapt to various environmental and scene processing requirements.

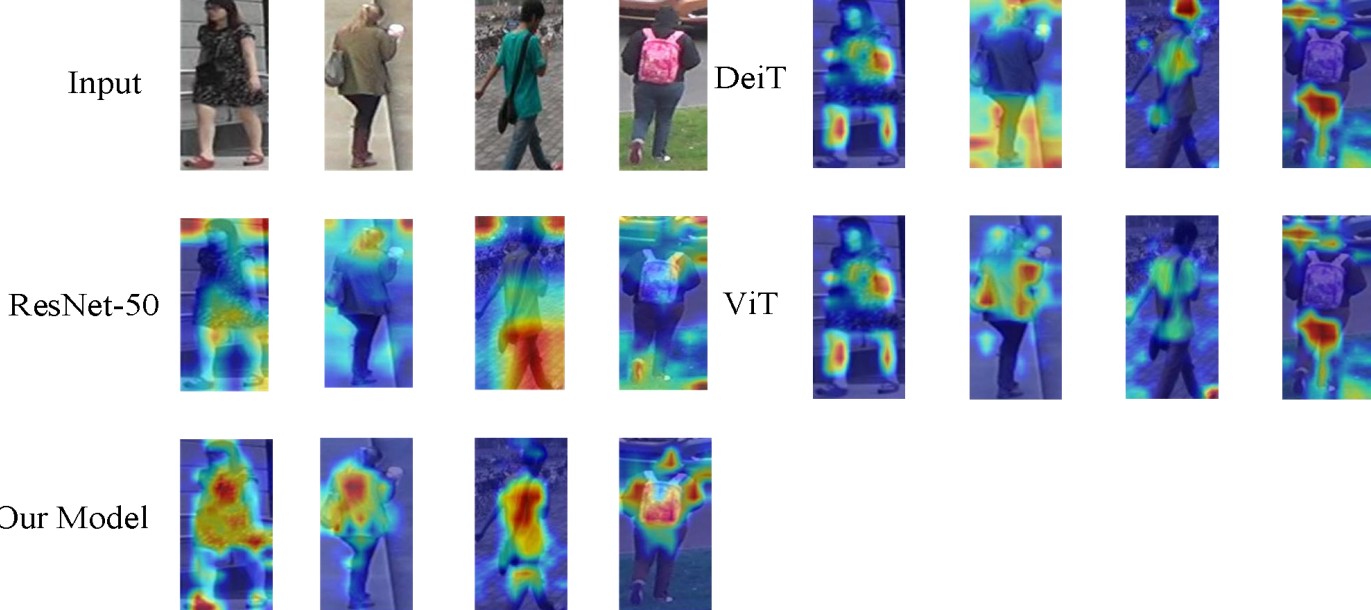

**Figure 5.** Visualization of feature maps based on different Re-ID models.

*4.8. Rank-List Visualization Analysis*

This section examines feature extraction and similarity matching for each probe person identification. We used the MHDNet, ViT, and ResNet-50 models to extract person features. Then, we applied a similarity matching algorithm to select the top 10 search results based on their similarity score. Correctly matched results are indicated in green boxes, while incorrectly matched results are highlighted in red boxes. The significant advantage of MHDNet in person recognition tasks is demonstrated by the in-depth analysis of query results for different test samples (numbered 1–4) presented in Figure 6.

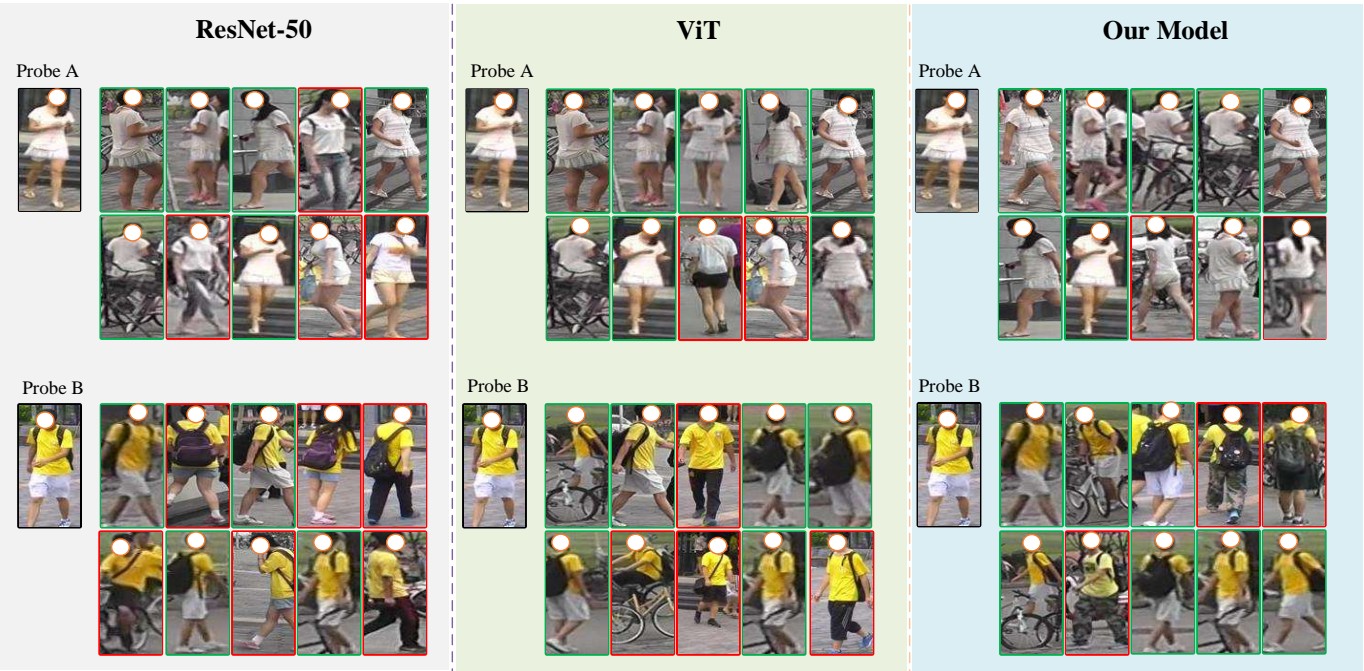

**Figure 6.** Top 10 visualization results of the rank-list Market-1501.

The inference results presented in Figure 6 demonstrate that MHDNet retrieves targets with the same ID rapidly and achieves better matching results than ResNet-50 and ViT models for each query. Furthermore, in complex cases, such as case B, MHDNet can correctly match the target and ignore other possible interfering factors, indicating that it has stronger robustness and accuracy and can perform well even in complex person recognition tasks.

## 5. Conclusions

This paper proposes an innovative hybrid multi-scale neural network architecture aimed at addressing the challenges in person re-identification (ReID) tasks. Firstly, we introduce a carefully designed spatial prior module that effectively extracts multi-scale features from different levels of network branches, capturing the diverse information present in pedestrian images. Subsequently, we construct a fusion module that bridges the gap between CNN and ViT backbone networks, leveraging a self-attention mechanism to learn global and local information of different granularities. This design successfully introduces image-related inductive biases into the architecture without altering the inherent structure of the ViT model, enabling the reconstruction of fine-grained multi-scale features crucial for accurate predictions. Through a series of comprehensive experimental validations, our dual-backbone model demonstrates superior performance in Re-ID tasks. Compared to currently well-designed vision transformers, our approach not only achieves comparable or better results but also does not significantly increase the number of parameters and computational complexity, thus achieving a good balance between performance

and efficiency. Looking ahead, we will continue to delve deeper into the pivotal role of multi-scale features in person re-identification tasks and strive to address the challenge of multi-scale alignment in multi-modal Re-ID. We anticipate contributing more innovative outcomes to the development of the person re-identification field through further research and optimization.

## 6. Privacy and Ethical Considerations

The proposed hybrid multi-scale neural network architecture, despite its promising performance in Re-ID tasks, necessitates a thorough examination of its privacy and ethical implications. The extraction of multi-scale features, albeit effective in enhancing ReID accuracy, introduces potential privacy risks associated with the handling of sensitive personal data. The integration of various networks further complicates these ethical considerations. Consequently, it is imperative to strike a balance between optimizing performance and safeguarding individual privacy, ensuring that the architecture adheres to ethical standards. Future research endeavors must prioritize the development of sophisticated privacy-enhancing techniques and ethical frameworks to facilitate the responsible and sustainable deployment of ReID technology within the academic community.

**Author Contributions:** Methodology, J.W. (Jinghui Wang); Software, J.W. (Jinghui Wang); Investigation, J.W. (Jinghui Wang); Supervision, J.W. (Jun Wang). All authors have read and agreed to the published version of the manuscript.

**Funding:** This research did not receive any external funding, and the article processing charges were borne by the author individually or by their affiliated institution.

**Data Availability Statement:** The data presented in this study are available on request from the corresponding author

**Conflicts of Interest:** The authors declare no conflict of interest.

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
