# Peer review of "MHDNet: A Multi-Scale Hybrid Deep Learning Model for Person Re-Identification"

_electronics, doi:10.3390/electronics13081435_

Round 1

Reviewer 1 Report

Comments and Suggestions for Authors

This paper proposes the MHDNet, which is a multi-scale hybrid deep learning model for person Re-ID. The proposed network is validated on two public datasets. Experimental results show that the proposed method outperforms state-of-the-arts.

1. The wording should be more explicit. For example, the “conventional operations” is ambiguous. More specific explanations would be helpful. Contribution 1: we explored…, while Contribution 2: we introduce… Please make this consistency.

2. In Figure 5, why only compare to ResNet-50? Please clarify this.

3. The hybrid concept should be more explicit. And other image processing tasks based on multi-scale features could be reviewed, such as Deep multi-scale features learning for distorted image quality assessment, Multiscale emotion representation learning for affective image recognition, etc.

4. How about multi-person Re-ID task? And there lacks the cross-dataset validation.

5. What is IEEE 2023 for? Please make it clearer.

Comments on the Quality of English Language

N/A

Author Response

尊敬的审稿人,请参阅附件

Reviewer 2 Report

Comments and Suggestions for Authors

The theme of the contribution is interesting, and relatively new. The structure of the document is adequate, and the text seems sufficiently scientific. What I am missing is a better emphasizing of benefits of your approach in comparison to other methods mentioned in the text and in conclusions (e.g., speed, model complexity, simplicity, accuracy, computational complexity,...).

Please please revise and expand the conclusions paragraph.

In the next work it would be appropriate realize more probes, not only two.

Author Response

尊敬的审稿人,请参阅附件。

Reviewer 3 Report

Comments and Suggestions for Authors

The authors present a work called MHDNet, a Multi-scale Hybrid Deep Learning Model for Person

Re-Identification. They begin with an introduction where they comment on the use that Re-ID has had in recent years and where current research on this topic has been derived.

Next, the authors review the related works, conveniently grouped into subsections. Next, the method used is explained exhaustively, and the variants with respect to the methods that are currently used.

The experiment is sufficient and quite well discussed. But the following observations are missing:

1- The ethical implications that could be had when using this system, in terms of violation of privacy, are not discussed.

2- The conclusions are quite short and general, for all the experimentation provided. This section should be expanded accordingly.

3- A study of future work could be added, so that this research team or another could continue the study from where it left off.

Author Response

尊敬的审稿人,请参阅附件。
